# Augmented Reality in Radiology for Education and Training—A Design Study

**DOI:** 10.3390/healthcare10040672

**Published:** 2022-04-02

**Authors:** Alexander Raith, Christoph Kamp, Christina Stoiber, Andreas Jakl, Markus Wagner

**Affiliations:** 1Department of Health Sciences—Radiologic Technology FH Campus Wien, University of Applied Sciences, 1100 Wien, Austria; alexander.raith@fh-campuswien.ac.at (A.R.); christoph.kamp@fh-campuswien.ac.at (C.K.); 2Institute of Creative Media Technologies, St. Pölten University of Applied Sciences, 3100 St. Pölten, Austria; andreas.jakl@fhstp.ac.at (A.J.); markus.wagner@fhstp.ac.at (M.W.)

**Keywords:** augmented reality, mixed reality, radiology, (patient) education, design study

## Abstract

Education is an important component of every healthcare system. Patients need to be educated about their planned procedures; healthcare professionals need to be trained in their respective profession. Both patient education and the training of healthcare professionals are often completed in person, which requires resources and is bound to certain times and places. Virtual educational environments can potentially save human and monetary resources, increase learner engagement, and enable users to learn according to their own schedules. This design study describes proofs of concept for two augmented reality-enabled (AR) educational tools, utilizing a Microsoft HoloLens head-mounted display. In the first use case, we demonstrate an AR application which could be used to educate cancer patients about their radiotherapy treatment and potentially reduce patient anxiety. The second use case demonstrates an AR training environment, which could complement the practical training of undergraduate radiography students. Two prototypes—VIPER, for patient education, and ARTUR for the training of radiography students—were developed and tested for viability and usability, both based on individual user tests. Both patient and student education were evaluated as viable and usable additions to conventional educational methods, despite being limited in terms of accessibility, usability, and fidelity. Suitable hardware is becoming more accessible and capable, and higher-fidelity holograms, better utilization of real-world objects, and more intuitive input methods could increase user immersion and acceptance of the technology.

## 1. Introduction

Augmented reality (AR) and virtual reality (VR) environments enable the widespread availability of simulated reality technologies. Various fields—e.g., entertainment and gaming [1], education [2], and sports [3]—have started to explore the advantages of these technologies, adopting them for their purposes. Additionally, many medical specialties are now also exploring the application of these tools for their respective fields [4,5,6]. As an example for the use of the technology in radiology, VR has the potential to improve the reading of radiological images [7]. Furthermore, VR and AR technologies have a great potential for radiology education and training [8]. In this paper, we present two use cases aimed at approaching some of the challenges in healthcare education utilizing the AR technology—VIPER and ARTUR—described in Section 4 and Section 5.

In the first use case, VIPER, we demonstrate the application of an AR prototype in the education of cancer patients to inform them about their treatment and minimize anxiety. Approximately 19.3 million people worldwide were diagnosed with cancer in 2020. The number of cancer cases is estimated to reach 28.4 million cases in 2040, a 47% rise compared to 2020 [9]. The main reasons for anxiety before the first irradiation include concerns about the side effects of the therapy, a process unknown to the patients, or about loneliness experienced in the treatment room during irradiation, the large treatment devices, and unusual noise [10]. Therefore, VIPER visualizes the treatment process with the help of a hologram intended to increase the radiotherapy patients’ understanding of their treatment. Therefore, it is of interest to know whether AR or VR devices are already used in the treatment and education of cancer patients, and whether VIPER has a positive influence on patient education from the perspective of health professionals in radiotherapy.

Proper patient positioning techniques are a crucial part of radiographers’ clinical routine. Therefore, the second use case demonstrates an interactive AR prototype for the education of undergraduate radiography students. The Augmented Reality Training Utility for Radiographers (ARTUR) enables the laboratory- and trainer-independent, AR-assisted training of patient positioning for plain radiography. Since the supply of readily available software is low and the initial costs associated with the purchase of AR hardware and software are rather high compared to traditional learning media, the added value of AR-supported training is often overlooked [11]. Today, most of the teaching and training of radiographic positioning is conducted either via simulation-based role-play training performed in laboratory exercises or during internships in a clinical setting. Supported by academic studies at the university, simulation-based training and clinical training are critical parts in the education of radiography students [12]. Virtual simulation-based training offers a variety of advantages compared to hands-on training. These advantages include, but are not limited to, its infinite reputability, portability, the possibility to track and analyze user progress, and the ease of wide distribution [13]. Like real-life simulation-based training, virtual simulation is already in use in a variety of healthcare applications. Virtual training environments can complement existing resources such as lectures, books, or e-learning resources. In addition, three-dimensional (3D) education environments can improve the contextualization of learning and increase the students’ engagement [14]. Moreover, virtual training environments enable learners to study without the presence of a teacher, which is needed while training in a laboratory or clinical setting. This enables students to study according to their own schedule and concentrate on their academic requirements [13].

This paper covers the following core contributions:We present two use cases—VIPER and ARTUR—showing the application of AR in two radiology application scenarios (see Section 4 for the VIPER use case and Section 5 for the ARTUR use case);We provide detailed descriptions of the problems, the design, the implementation, as well as the validation of the AR prototypes. We followed the well-known nested model for visualization and validation along with the design methodology of [15]. In Figure 1, we illustrate the use methods in detail;We discuss and generalize the results of both design studies in Section 6, showing that AR has a great potential in education as well as radiology training.

## 2. Methodology

Our main research aims are the following:What is the impression of health professionals in radiology regarding the utility of AR?How can virtual learning environments support radiographers in understanding radiological procedures?

To answer these question, we followed the nested model for visualization design and validation [16], which is illustrated in Figure 1. This unified approach splits visualization design into four levels, in combination with corresponding evaluation methods, to evaluate the results at each level:Domain problem and data characterization;Operation and data type abstraction;Visual encoding and interaction design;Algorithm design and implementation.

Since these are nested levels, the output of the upstream level is the input of the downstream level. First, we started with a problem characterization and abstraction based on the design study methodology [15], which brings us to the first level (domain problem and data characterization) of the nested model [16]. From there, we worked inwards along the nested model for visualization design and validation. To perform the problem characterization and abstraction, we followed a threefold qualitative research approach, which consists of a systematic literature survey (in the VIPER (Section 4) and ARTUR (Section 5) use cases) and a focus group [17] with domain experts [17] (ARTUR use case, Section 5).

To provide a good overview of different aspects of AR in medical settings, we started with a literature survey (see Section 3) describing both areas, as well as a general view for AR. In detail, we focused our search on reputable journals, starting in 2010, containing the following keyword combinations: radiographic positioning training; radiographic positioning virtual training; virtual training radiographer; augmented reality radiography; augmented reality training radiology; augmented reality training healthcare; anxiety in radiotherapy; information procurement radiotherapy; anxiety cancer patients; information procurement cancer patients; patient education; virtual reality cancer/radiotherapy patient education; augmented reality cancer/radiotherapy patient information; virtual reality cancer/radiotherapy patient education. Based on this search, we found articles and studies dealing with the evaluation of either hands-on or virtual training of radiographic positioning and radiation therapy, and identified the preferred and trustworthy information sources of cancer patients. Since no AR training applications for radiographic positioning could be found, articles dealing with AR training for other healthcare professions were also considered.

In the next steps, we started with the visualization and interaction design, followed by the algorithm design and implementation, based on a user-centered design process [18]. For this purpose, we produced functional prototypes [19]. By doing this, we fulfilled the third (visual encoding and interaction design) and fourth (algorithm design) levels of the nested model. Additionally, we performed user studies to evaluate the usability [20] of the designed AR systems and to generalize the results and insights (see Section 4.3 and Section 5.4).

To validate the usability and supportiveness of AR in medical settings, we performed two case studies that are described in detail in this article (see Section 4 and Section 5). We followed a problem-driven approach in the context of real-world problems.

The research methods for both prototypes (VIPER and ARTUR) were reviewed by the Lower Austrian Ethics Committee and were not objected to. Both projects were approved based on the reviewed methods. Since 2 different groups of subjects were involved, different procedures were submitted.

## 3. Related Work and Background

In this Section, we describe the paper’s related work in detail and from different points of view, structured along the core topics: (1) AR in education and training; (2) AR/VR radiotherapy patient education; and (3) AR in radiology training.

### 3.1. AR in Education and Training

The commonly used definition of mixed reality (MR) by Milgram and Kishino [21] states that MR merges real and virtual objects into a “virtuality continuum”. While VR describes a purely virtual environment, the terms MR and AR are often used interchangeably.

AR can be based on different technologies: common systems are head-mounted displays (HMD) with see-through displays that only render the additional holograms, or smartphone-/tablet-enabled devices that augment a real-time video feed captured through a camera. The second option has a greater potential market reach, as it is integrated in all recent Apple iOS based devices (https://developer.apple.com/documentation/arkit/verifying_device_support_and_user_permission, accessed on 22 November 2021) as well as most Google Android based devices (https://developers.google.com/ar/devices, accessed on 22 November 2021). This is important for learning and training scenarios where users are expected to use their own devices. A good example is the EPAR project [22], which focuses on patient education in the waiting room before eye surgery. In this scenario, patients have the option of augmenting a printed information sheet with an interactive visualization for improved health literacy.

However, the HMD variant is more immersive, as the 3D holograms are embedded directly into the user’s field of view (FOV), and not just visible on a 2D device screen. The added immersion can have advantages for complex visualizations, but both variants (HMD and smartphone-based) are effective for learning, and are comparable with regard to their educational effects [23]. The Microsoft HoloLens (https://docs.microsoft.com/en-us/hololens/hololens2-options, accessed on 30 April 2021) is a good mix of general availability for enterprise customers combined with a specific design for immersive AR. It includes hand tracking for controller-free interaction, as well as a time-of-flight depth sensor and spatial mapping for robust anchoring of holograms to real-world places. It runs the Windows Holographic Operating System (a variant of Windows 10), which allows stand-alone and untethered operation as well as integration into the organization’s device management for additional login and application security. Industry certificates allow HoloLens usage in clean rooms (ISO 14644-1 Class 5) as well as in hazardous locations.

With available commercial AR devices, the uptake in both commercial applications and research for industry use cases [24], as well as in medical treatment and education scenarios, is growing.

The number of recent scientific publications indicates that AR is often used in training. According to the conducted literature survey, 29% of related publications target higher education [25]. Within the medical field, most of the AR prototypes focus on support and education for doctors and medical students. Ref. [26] supports doctors in planning and completing oral medical procedures, while [27] helps mentors and mentees with the augmentation of surgical tele-monitoring. Further educational aspects in this field include AR edutainment systems for learning bone anatomy [28], AR for medical diagnosis [29], or rehabilitation [30].

### 3.2. AR/VR Radiotherapy Patient Education

Educating patients on procedures they may undergo is important in order to reduce anxiety and help them feel adequately prepared for their treatment [31]. Virtual patient education tools may also support the patient information process [32,33].

To date, only one radiation therapy patient education program has been found in the literature. The Virtual Environment for Radiotherapy Training (VERT) includes models of a treatment room, a linear accelerator, a treatment machine, various treatment aids, a treatment couch, and furniture. To visualize a patient’s radiotherapeutic treatment method, the oncologist or radiographer can import the patient’s computed tomography (CT) dataset and irradiation plan into VERT and position the patient on the treatment couch. The movements and noises of almost all the devices in the virtual treatment room correspond to reality, and the individual irradiation plan of each patient (including anatomy, organs at risk, dose distribution, and target area) can be presented in 3D (on a large screen, around 4×3 m), which helps to explain the complex and complicated procedure. A number of studies [34,35,36] have explored which information is important for radiology patients while going through the process, and whether the VERT system increased the patients’ and their relatives’ knowledge of radiotherapy planning and delivery [34]. The results emphasized the positive influence of VERT information on patients’ understanding of radiotherapy. Using the 3D representation of the treatment and the equipment served to lower the patients’ anxiety level before the first irradiation. A further study by Stewart-Lord et al. [35] examined the perception of VERT among prostate cancer patients based on an innovative information delivery tool. The results of this study also demonstrated a positive response to the VERT for prostate cancer patients. Another study by Hansen et al. [36] investigated whether patient education with VERT had an influence on residual setup errors during irradiation, and whether the number of repositionings decreased. The VERT information session and the training before treatment resulted in a significant reduction of residual rotational errors in the intervention group compared to the control group, but no significant differences were identified in the number of repositionings. This result showed the importance of teaching patients about the necessity of correct positioning to improve positioning prior to treatments.

In summary, these studies revealed that AR/VR technology has the potential to be a powerful education tool for radiotherapy patients. However, there are currently no studies in which the explaining medical staff can assess the influence of these informative methods on the information conversation. Moreover, both technologies can also have great benefits for professional radiology training. In the following, we present the current state of the art of AR in the training of radiological procedures.

### 3.3. AR in Radiology Training

The training of radiographic positioning is a crucial part of the clinical education of radiography students. Even though contact to real-life patients provides students with optimal conditions to develop their skills, clinical environments are not able to offer students safe learning surroundings. In addition, patients are entitled to receive the best possible and safest treatment available, which makes the use of simulated learning environments in student education essential [37].

Clinical education, where students perform radiographic examinations on patients under the supervision of registered radiographers in a clinical setting, is recognized as an important core element of students’ education in radiography [12,38,39]. Clinical placements are not universally applicable and students often find the transition from academic studies to clinical practice challenging, which hinders the formation of a clear link between theory and practice [40,41,42].

Hands-on training in a laboratory setting can provide a safe and effective environment to transfer theoretically acquired knowledge into practice. These laboratories need to be equipped with radiographic equipment, all of which adds to the expenses necessary to realize hands-on simulation training such as role-play scenarios and phantom studies. Virtual simulation enables students to review the content at any time, without being bound to the laboratory [14].

Shanahan [38] researched the perspective of radiography students on using a virtual radiography simulation. The pilot study introduced a virtual radiography simulation in the form of the computer-based software Projection-VR, now known as Shaderware. The simulation acted as a support for the development of pre-clinical technical skills in the second year of an undergraduate radiography program. To evaluate the students’ perceived confidence and skill development, and possible usability issues the students might encounter, the study utilized a mixed questionnaire containing Likert-scale and open-ended questions. Educators might also underestimate the level of guidance and training needed by the users before the virtual simulation can be utilized efficiently [38].

Being computer-based and controlled through a keyboard and mouse, Shaderware offers relatively low fidelity. To research the impact of a more immersive, higher-fidelity simulation, and to compare it to computer-based software, Sapkaroski et al. [43] developed a virtual reality-based software and integrated it into the curriculum of a radiography program. The virtual reality environment featured dynamic patient interaction with voice recognition in simulated clinical scenarios [43].

In line with other studies, the results of the questionnaires indicate that the use of virtual simulation aids students in their training of radiographic procedures [44,45].

At the time of writing, no AR-enabled simulation environments for the training of radiographers were available. However, the feasibility of AR simulation in healthcare education has been proven by a study testing an AR cardiopulmonary resuscitation training system [46].

## 4. Use Case: VIPER

VIPER is an AR prototype aiming to support cancer patients by minimizing their anxiety before their first irradiation and, at the same time, to support oncologists and radiographers by providing a dependable source of information. In the following, we elaborate on the requirements derived from the related work, the design and implementation, as well as on the validation of the system.

### 4.1. Problem Characterization

Once a cancer diagnosis has been confirmed, there are several treatment options. The most common therapies consist of one or more of these options: surgery, chemotherapy, hormone therapy, immunotherapy, and radiotherapy.

Radiotherapy is the use of radiation (i.e., gamma rays, protons, X-rays, or electrons) to destroy cancer cells or to damage them in order to prevent their further growth. This treatment is localized, which means that only the radiated part of the body is affected. Cancer cells should begin to die within days or weeks after the treatment starts, but this effect also continues for weeks or months after the treatment has been finished. The disadvantage of this treatment option is that radiation can also damage healthy cells and side effects may occur. The possible occurrence of side effects, the uncertain prognosis, and the use of large medical machines (linear accelerators), as well as the fact that the exact course of a radiation session is unknown, can lead to great anxiety and concern among patients. Some studies conclude [47,48,49,50,51,52] that anxiety levels are highest before a first radiotherapy treatment session, while good patient-oriented education can reduce the extent of anxiety. The VIPER system presented here is intended to support health professionals in their patient education.

So far, previous studies [52,53,54] have confirmed the need to prepare patients for the exact details of their upcoming radiotherapy. Based on these studies, we derived the following requirements and conditions for our application regarding the information radiotherapy patients need to reduce their anxiety:Before the therapy starts, possible side effects, the treatment procedure, and the prognosis are of particular interest;The favored information sources are in written or verbal forms, but internet sources are also used, although their quality is sometimes doubtful. Therefore, it is important to use and offer tailored and diverse forms of information.

Since side effects and prognoses are very individual, and since we aimed to develop a tool that would be useful for the majority of patients, it was decided to show the treatment process virtually, since it is very similar for most patients in radiotherapy. VIPER, the virtual patient education tool, explains the exact process of irradiation. To date, a virtual patient information tool in radiotherapy is available only through the VERT system. Since this system is locally bound and demands a lot of space, a requirement for the development of VIPER was to enable mobility, while at the same time ensuring easy handling by the users. Easy handling means that the preparation for the use of VIPER is not time-consuming and, in addition, the provision of the tool remains user-friendly (i.e., there are only a small number of interactions before the program starts). Cancer patients may suffer from various physical limitations due to their disease and previous therapies, including movement, motor, sensory, speech, and visual impairments. Owing to these possible limitations of the patients, interaction with the program via gestures or speech is best avoided; thus, we opted for an education via AR instead of VR. On the one hand, this is to ensure that the probability of motion sickness is minimized and, on the other hand, that a problem-free optical connection between the doctor and the patient is still possible during use.

### 4.2. Design and Implementation

VIPER models the most important components in an irradiation room. In the following scenario, we describe the VIPER system in detail (see Figure 2): to minimize the interaction of the patients themselves for reasons already mentioned, the program should be started before the patient puts the HoloLens 1 on. From this point on, the course of an irradiation is presented in an automated way, and no further interactions are necessary. At the beginning of the sequence, the patient waits for a call to enter the irradiation room and place themself on the treatment table. Subsequently, the treatment table moves close to the irradiation position. Based on the position of the laser and the skin markers (which have to be brought into alignment), the patient lies in a crooked position and a correction of their position is carried out. Then, the gantry moves to its initial position and, shortly thereafter, the irradiation starts. Care was taken to make the irradiation duration of the animation match the duration of the actual irradiation. After the irradiation process is completed, the gantry device moves back to its original position and the table descends so that the patient can get up and leave the room. After the modeling and the animation were completed, VIPER was provided with sound. In addition, all the animated events were explained and the movements of the linear accelerator and the treatment table were equipped with sound as well. The duration of the whole information animation was 2:45 min. Apart from starting VIPER (which can be completed by the oncologist or the radiographer), no further interactions were necessary. In this pilot project, the procedure of irradiation of the prostate is presented. The reason for this is that this irradiation is relatively easy to explain and present and, moreover, this carcinoma is the most common in the male sex. Other procedures, such as the irradiation of a breast carcinoma (the most common carcinoma in females), which has somewhat more complex procedures for virtual representation, are to be developed in the future.

Professional representation of the course of radiotherapy required models of the most important components in an irradiation room. Some models were available for developers free of charge (some of which we could use just as they were offered, while others had to be modified), and others were created by the authors. The application Autodesk Maya 2017 was used to create and modify the necessary models. The basic model for the patient was freely available for developers but had to be adapted intensively. In addition to the appropriate position of the arms and legs for irradiation, the skin markings were also implemented. The model for the linear accelerator was also available online and free of charge for developers. Positioning devices (head pillow and knee pad) were added, and the position of the in-room monitors was changed in order to allow for ideal camera positioning. In addition, individual components of the treatment table were modified to ensure optimal animation, and a laser system was added schematically for the realistic presentation of the radiotherapy course and to explain the significance of accurate positioning. In fact, the laser beams are not clearly visible in the room, but this representation promotes intelligibility. This laser system was positioned on both sides next to the treatment table. The scene was animated after all the models were completed. The application Unity 5.4.0 was used to perform scene animations. The audio files for the explanation of the irradiation process were recorded and subsequently reworked with the open-source application Audacity. The sounds for the device movements were recorded with a microphone in a real-life linear accelerator room. In the end, the animations and soundtracks were brought into correlation with each other to complete VIPER. To simplify the upcoming utility test regarding the use of this patient education method, the program was fitted with additional scripts to start and restart the animation with user input (gestures). In order to minimize the necessary level of user interaction, the animation ran in a continuous loop.

### 4.3. Validation and Reflection

We conducted a user study to evaluate the prototype. The focus was on the potential benefits of the VIPER system in the patient education process. Therefore, experts who perform this patient education were interviewed; the influence on the actual change of the anxiety level was not evaluated. For this, a survey with 22 radiotherapy experts who are confronted with patient education in their daily work was conducted. A total of 16 participants were radiographers and 6 were medical staff (4 specialists for radiooncology and 2 assistant doctors). The first part of the questionnaire contained general questions about the person, such as age, gender, and work experience, as well as prior knowledge of or past experiences with AR/VR devices. The participants were asked to answer this part before testing VIPER.

The test itself took place in a quiet room with a table where the hologram was placed. Before the program was started, the basic structure and function of the HoloLens were explained. Afterwards, every participant was supported during the positioning of the HoloLens on their head to ensure an optimal position and, thereby, guarantee an optimal projection of the hologram. After the correct fit of the glasses was verified, the participant started the patient education program VIPER.

After patient information was active and the animation was still running, the participants could move freely in the room in order to allow different viewing angles of the three-dimensional projection of the treatment process. The entire VIPER patient information process could be repeated as often as required to acquire every bit of information for a conscientious evaluation. For this, the second part of the questionnaire was handed out. This part consisted of nine items, seven of which were statements that could be assessed on a Likert scale from 1 to 5 (1: does not apply at all, 2: does rather not apply, 3: partially applies, 4: largely applies, and 5: fully applies). The other two items were open questions. Since the questioning of the experts took place during their daily clinical routine, care was taken to keep the questions brief and concise. A high number of open or in-depth questions was therefore avoided. This type of survey was intended to achieve a high response rate.

Additionally, the participants were asked whether they believed that digital media would influence patient education in the future. With a mean score of 4.09 (SD 0.85) obtained from the responses of all participants, it is assumed that media technology will have a substantial impact on patient education. Radiographers are more likely to be convinced than physicians (4.38 to 3.33).

Patient education talks in radiotherapy usually take a lot of time (about an hour) because there can be many questions and problems of understanding on the part of the patients due to the complexity of this therapy. As one of the main goals in the development of VIPER was to support healthcare professionals in the education process, subjective questions (potential to save time and increase understanding) were specifically asked in this proof-of-concept project.

As is evident from the distribution of the answers, a big benefit is seen in VIPER for increasing the understanding of the course of irradiation in the cancer patients. The average value of all responses was 4.05 (radiographers 4.38 and medical staff 3.17), while the median was 4 and the IQR 1. While the benefit of VIPER was recognized, the health professionals did not expect or were not sure that an application of this program would additionally save time (mean 3.05, median 3, IQR 2). No participant answered this question with “fully applies”, while 9 answered with “partly applies” or “largely applies”, and 10 with “does rather not apply”. A summary of the described average responses can be found in Figure 3.

In addition, the participants were asked about the general usefulness of the VIPER system and whether they would recommend using it. There was a considerable difference in the answers between radiographers and medical staff. While the radiographers would appreciate support through VIPER in patient education (4.19), the physicians were not fully convinced of the benefits of the tool (2.67). The median value was 4 overall and the IQR 2. In absolute values, this means that 13 (81.3%) radiographers answered the question concerning utility with “largely applies” or “fully applies”, and none with “does not apply at all” or “does rather not apply”. Owing to the large number of radiographers in the sample, the usefulness of VIPER was assessed positively on average (3.77).

In the final closed question, the author was interested in whether the participants would recommend the VIPER prototype to their colleagues. Six radiographers (and one doctor) answered with “fully applies”, while five radiographers and none of the doctors answered with “largely applies” and four radiographers (and one doctor) answered with “partially applies”. That means that 77.2% of all participants would recommend the program at least partially. The rest of the sample answered either with “does rather not apply” (two doctors, one radiologic technologist) or “does not apply at all” (two doctors). The average response to the question of the recommendation of VIPER was, for radiographers, 4.00, and for the medical staff, 2.33, leading to an overall result of 3.55 with a median value of 4 and an IQR of 2 (Figure 3).

### 4.4. Limitations and Discussion

It is noticeable that the medical staff generally answered with lower values compared to the radiographers. On average, the doctors gave 0.94-points-lower answers to the questions than the radiographers. It is important to note that the doctors who participated in the evaluation of VIPER were, on average, older than the radiographers. Morris and Venkatesh [55] reported that their study had also shown that age-related technical acceptance was more pronounced when the information presentation was given in a new way (such as holographic presentation with VIPER). The results of their study also suggest that there are clear age differences when it comes to attaching importance to various factors in technology adoption and use. Whether this is the reason for the lower evaluation of the questions in our study cannot be proven, but it should be considered as a possibility and can be the subject of further investigations. Finally, it should be emphasized that VIPER left a positive impression among the experts, and many of them would recommend this program to a colleague (mean score 3.55).

One limitation was the small number of participants. However, since radiooncology is only a small part of radiology and not all radiooncologists perform patient education, the number of persons who participated was well within expectations. Although the number of participants in the patient education studies using *VERT* was higher, it is worth mentioning that these studies have conducted a survey among cancer patients whose number is many times higher than those of the employees of a radiooncology department.

Another limitation was the place where the survey was conducted. On the one hand, only persons from one radiooncology department were asked to participate, which makes generalizations and comparisons difficult. On the other hand, the survey was conducted during the clinical routine in the department. Moreover, due to time pressure, patient care, and other tasks, the participants often had limited time to deal intensively with VIPER and to assess the full scope of this information tool or to think about the consequences of its use.

## 5. Use Case: ARTUR

The goal of developing a prototype for an Augmented Reality Training Ulitity for Radiography (ARTUR) was to enable the laboratory- and trainer-independent, AR-assisted training of patient positioning for plain radiography. The design of ARTUR aimed to provide users with life-size reference holograms of radiographic positions. Users are then able to use the hologram as a positioning reference while bringing a training partner, who acts as a test patient, into the correct position required for plain radiographic imaging. The prototype includes reference holograms for three radiographic positions: two projections of the scaphoid bone and one projection of both hands. In the following, we present the problem characterization and requirements as well as the design and implementation of ARTUR. Additionally, we present the results of the heuristic evaluation, accomplished through individual semi-structured expert interviews.

### 5.1. Background

Radiographers are required to display high levels of accuracy when positioning patients, as even slight over- or under-rotations or misalignments can result in poor image quality and complicate the diagnosis of a radiographic image. In addition, radiographers are obliged to protect patients from unnecessary radiation exposure, which requires radiographic images to ideally be acquired on the first try [56].

Standard radiographic positioning allows radiologists to describe findings on radiographs in relation to anatomic structures. Standard positions ensure comparability between multiple instances of radiographic images. Berlin [57] argued that a radiologist’s accuracy in reporting radiographs cannot be greater than the quality of the radiographs presented for interpretation. Proper radiographic positioning is essential for high-quality radiographs.

Practical training of radiographic positioning in clinical or laboratory settings represents a crucial part of the education of radiographers. Since hands-on training on real radiographic equipment in the presence of a trainer or clinical placements in a hospital demands a lot of resources, alternatives are sought [14]. Virtual training applications have shown to provide effects similar to hands-on training while conserving financial and human resources [58].

### 5.2. Problem Characterization

Since the conventional training of radiographic positioning has its limitations [40,41,42], an AR-enabled prototype for a virtual training environment was developed. AR was chosen as a technology to close the gap between theory and practice, which was observed to present an issue for radiography students [59]. With the use of AR and the projection of life-size holograms into the user’s surroundings, a mixed approach to digital and analogue training was chosen. While users were guided by a virtual application, they still had to apply their theoretical knowledge to real-life training partners.

To test the feasibility of a life-size AR-assisted guidance system for radiographic positioning, this study aimed to test the usability of such an application on test users with varying levels of experience with AR. Only if the concept is deemed to be usable are further developments and surveys into the learning effect of such an application sensible.

To specify the user requirements of ARTUR, a focus group interview with four teaching radiographers was conducted. The focus group discussed the possibilities and requirements of AR-assisted practical training in the education of undergraduate radiographers. In accordance with current literature on the topic of simulation-based training in healthcare education, all participants of the focus group recognized the importance of practical training in learning radiographic positioning [37,39,40]. They also stated that new and innovative tools promote student engagement and motivation [14]. The capability of AR to provide reproducible training environments independent of space and time was also mentioned by several participants. In terms of further advantages of virtual simulation, potential cost and time savings were mentioned.

Regarding visual quality, the importance of fidelity was emphasized. Realistic visuals improve immersion and learner engagement. Furthermore, the process of radiographic positioning makes use of anatomical landmarks, which need to be visible in the simulation. The application should preferably use high-fidelity 3D models that resemble the original body parts as closely as possible. Hand gestures, as an input method to interact with holograms, were universally requested by the focus group participants. A speech interface enabling users to interact with voice commands was also mentioned. User feedback on interactions is crucial for AR applications. The focus group had several suggestions with varying degrees of feasibility, including haptic feedback through a vibrating headset and audio-visual feedback. Based on the focus group discussion, we can present the following requirements: sufficient visual quality, the possibility for user input, and feedback on the user input. The feedback on the user’s input should include visual and audible feedback.

Visual Feedback: Holograms of a body part to be positioned and visual representation of parameters set at the X-ray tube, including central beam position, collimation size, detector size, X-ray tube position and angle, exposition parameters, and the resulting radiographic image;Audible Feedback: Text-to-speech output;Visual Quality: Realistic representation of body parts and radiographic equipment;User Input: Hand gestures and speech input;Input Feedback: Haptic, visual, and audible feedback.

For the design of ARTUR, a hierarchical task analysis was conducted. Task analysis is a process which helps with understanding how users perform tasks and achieve their goals. What actions do users perform to achieve their goals? What is the personal, social, and cultural background of the users? How are users influenced by their environment, and how does the users’ knowledge and previous experience influence their workflow? This supports the identification of the tasks the application must support and helps with the development and revision of new features. The task analysis was performed early in the development process, before any design work was conducted [60]. Hierarchical task analysis is focused on dividing a high-level task into smaller subtasks. The goal of this high-level task decomposition is to display the overall structure of the main user tasks. To carry out a task analysis, several steps were taken. The main task was identified and broken down into seven subtasks. A layered diagram gives an overview of the subtasks [60]. The main user task was defined as completing a simulation of an entire radiographic positioning procedure. The main task was then divided into seven subtasks, with each subtask representing a single user interaction.

The seven subtasks were identified as follows: (1) choose a body part by selecting a radiographic procedure via the Air Tap gesture in the menu; (2) position the body part on the training surface by grabbing the hologram with the Air Tap gesture and moving the hand; (3) rotate the body part to comfortably position the test patient into the hologram by grabbing a rotation point on the hologram’s bounding box via the Air Tap gesture and moving the user’s hand; (4) resize the hologram to match the size of the test patient by grabbing any corner of the hologram’s bounding box with the Air Tap gesture and moving the hand; (5) position the test patient as shown by the hologram, with the previously positioned hologram acting as a reference; (6) display the central beam by selecting the central beam button in the menu using the Air Tap gesture; (7) display additional examination parameters by selecting the corresponding button in the menu via the Air Tap gesture. The individual user tasks and their corresponding user interactions are illustrated in a flow chart in Figure 4.

### 5.3. Design and Implementation

Based on the focus group discussion and the task analysis, a working prototype of ARTUR was developed for the Microsoft HoloLens 1 with the Unity 3D 2018.4 development platform, utilizing the Microsoft Mixed Reality Toolkit (MRTK) 2.3.0 for Unity.

To provide the application with high-fidelity holograms, real body parts, positioned for radiographic procedures, were scanned with an Artec Eva structured light 3D scanner. For the prototype of ARTUR, three body parts were scanned and processed into three 3D models to be used as AR holograms. An example of a 3D-scanned arm in position for two radiographs of the scaphoid bone is shown in Figure 5.

The menu consisting of floating buttons was configured to follow the user’s gaze and to always face the user.

Two scripts were necessary to allow user manipulation of the holograms. A Near Interaction Grabbable script enabled the user to interact with the object by gazing at it and performing an Air Tap gesture. To allow rotation and resizing of the body parts, a Manipulation Handler and a Bounding Box script were used. The Manipulation Handler manages the permitted types of manipulation by the user—in this case, users can move, rotate, and scale objects.

To give the hologram physical attributes and to allow interactions with real-world objects such as tabletops, a Box Collider script and a Spatial Mapping Collider script were added to the object. The Box Collider sets the boundaries of the Bounding Box as borders of the object and gives the object rigid attributes. The Spatial Mapping Collider, which is linked to the size of the Box Collider, enables the hologram to collide with real-world surfaces, which enables users to set the hologram on a tabletop or any other surface of their choice.

To simulate the collimation light and the central beam crosshair of an X-ray machine, a spotlight was set as a child of each hologram. The central beam crosshair was positioned on the holograms according to the guidelines for radiographic imaging [61]. The collimation size was adapted to the size of the scanned body parts. Since the spotlights were set as children of the models, the collimation size adapted to resized models. The user interface and a hologram, as seen from the user’s FOV, is shown in Figure 6.

Technical parameters for the examination, such as recommended detector size, voltage, or current, are conveyed through text on a virtual board next to the menu. To play an audio version of the description, users can Air-Tap the panel.

### 5.4. Validation and Reflection

To evaluate the usability and feasibility of ARTUR, a heuristic evaluation approach, as described by Nielsen and Molich [62], was chosen. Three user tests, followed by semi-structured expert interviews, were conducted [63]. Prior to the interview and directed by a testing guideline, each test user performed an independent test run of the application. Test users who were inexperienced with AR received an initial explanation of the HoloLens and its input methods. Following this briefing, all test users completed the tasks provided by the testing guideline without further intervention. The testing guideline was designed to simulate a training session of two radiographic positions.

Prospective interview participants were identified using an expert definition by Bogner and Menz [64], who stated that experts have technical process-oriented and interpretative knowledge of topics relevant for their specific professional activity.

#### 5.4.1. Design and Procedure

Following the user tests, which were executed individually with each of the participants, the interviews with the experts were conducted in a one-on-one setting in March and May 2020. Before conducting the interviews, all participants signed a form of consent. The interviews were conducted according to an interview guideline. The guideline was designed to address topics relevant to the design and functional features of ARTUR. Including the introduction and discussion of the participants’ questions, each interview lasted approximately 45 min.

#### 5.4.2. Apparatus and Material

To allow for an analysis of the interviews, audio recordings of all three interviews were made and transcribed [65] (p. 83). The transcripts were thematically categorized by statements relevant for the feasibility and usability of the application. To allow for a systematic evaluation, the functional requirements set during the design process of ARTUR were used as a framework for a comparison of the interview findings. Testimonies indicating either common ground or major disagreements between the experts were also highlighted, categorized, and included in the findings.

#### 5.4.3. Results

To measure the impact of issues found by the test users, each identified usability problem was rated according to its severity. The severity ratings for usability problems by Nielsen [66] indicate the need of resources to be allocated to fix usability issues. Since ARTUR was limited to run on first-generation Microsoft HoloLens hardware, hardware-related issues were given a severity rating of zero. While being able to decrease usability, hardware-related issues could not be fixed but only worked around. Since an updated version of the software is currently being tested on the HoloLens 2 hardware, several improvements due to the updated hardware could already be identified. The lighter headset improves user comfort, and the larger FOV improves user orientation and prevents the users from “losing” the holograms in their environment. The greatest advantage is offered by the improved hand tracking provided by the HoloLens 2. Directly grabbing and manipulating the holograms with both hands is deemed to be more intuitive for all users. Any software-related usability problems directly or indirectly caused by hardware issues received their own severity rating. All identified issues and their respective usability ratings are illustrated in Table 1. Overall, users found the design and usability of the application suitable for its purpose. During the evaluation, no major usability issues or usability catastrophes were identified. All users identified three major hardware-related usability issues, which could not be addressed on first-generation HoloLens hardware. The issues identified were: lack of comfort while wearing the headset, the seriously limited FOV provided by the first-generation HoloLens, and the input methods limited to a few gestures only. Regarding the menu system provided by the application, the users suggested offering a better visual distinction between interactive and non-interactive menu elements. A lack of distinction between interactive and non-interactive menu elements is a frequent issue that is encountered by each user every time the menu is in use. Since manipulating non-interactive elements does not trigger any actions, the impact can be rated as low. The same applies to the persistence of the problem. Once users are aware of the differences between buttons and labels, usability is not restricted. It can, therefore, be rated as a cosmetic problem which should only be fixed when resources are available. As a cosmetic issue, it is rated as a “one” on the severity scale. One test user reported a usability issue with transparent box colliders which were bigger than the holograms. This did not allow for the manipulation of buttons visible behind the box collider. Users not being able to manipulate visible objects because they are occluded by transparent collider boxes is a frequently encountered issue. Impeding users from interacting with the application’s menu or holograms results in a usability problem which is more severe than a merely cosmetic issue. Users can overcome this issue by moving holograms or the menu in their space, which makes the issue non-persistent. It can be classified as a minor usability issue with a rating of “two” on the severity scale. To enhance the user experience, improved distinction between interactive UI elements and non-interactive labels as well as modified hologram colliders should be implemented in the next iteration of ARTUR. General limitations such as the limited FOV, lack of comfort, and non-intuitive gestures can be attributed to hardware limitations of the first-generation HoloLens.

### 5.5. Discussion and Limitations

Although virtual simulation environments for the training of plain radiographic procedures do exist and the impact of virtual simulation has been researched, no research of the AR-supported training of plain radiographic positioning had been conducted at the time of writing. In accordance with the current literature, the focus group recognized virtual training environments and AR as a promising technology to enable time- and cost-effective independent learning. The focus group agreed on virtual simulation representing additional tools for education, complementing traditional methods such as hands-on training, lectures, and clinical practice. They had high expectations for the capability of AR to boost learner engagement.

The evaluation of ARTUR showed that AR applications can be utilized by users with any level of experience with AR. The evaluation has also shown that many limitations are caused by last-generation hardware. While all test users were able to navigate the application, none described the interaction with the HoloLens 1 as intuitive. This suggests a need for initial training in users who are new to AR. Porting the application to the second-generation HoloLens could solve some hardware-related issues and further improve the usability of ARTUR. Further limitations to the broad deployment of AR-supported training are the limited availability of current AR hardware and the high cost of purchase for said hardware. This limits the potential user base to educational institutions or healthcare providers. With a decrease in hardware costs and a wider distribution of AR headsets, true time- and space-independent training of radiographic positioning could become a reality.

A major limitation of this case study was the limited number of expert interviews. Due to the COVID-19 situation in early 2020, it was impossible to carry out in-person user tests with healthcare professionals or students on a large scale. This led to the extension of the expert criteria to include AR/VR experts and to a reduction in the number of interviews. Further limitations can be attributed to the utilization of only one iteration of a user-centered design circle. As future prospects for this project, further iterations of the user-centered design circle should be applied. Following an implementation of requested features and suggested fixes, another evaluation by an extended group of test users could be carried out.

## 6. Conclusions and Future Perspectives

In this paper, we presented two use cases (VIPER and ARTUR) for AR in medicine, especially for radiography and radiotherapy. Both were developed and tested with the Microsoft HoloLens 1, but can be easily transferred to the HoloLens 2. The development was conducted in a user-centered design approach, as can be seen in Figure 1. To quantify the impact of virtual learning environments on the education of its users, this paper utilized recent literature to analyze the current state of the art in virtual training. Several studies indicate that simulation-based training helps users as well as professionals in developing practical skills and in gaining knowledge in the applied field.

The development of the prototypes has shown that AR can be used to visualize rather complicated procedures such as the irradiation procedure in radiation therapy or the exact patient positioning for radiographic procedures. These visual aids in the form of holograms can be used to complement traditional forms of knowledge transfer.

After health professionals evaluated the two prototypes developed for this paper, the following impressions about the utility of AR in healthcare education could be summarized. The test users assumed that media technology, including AR, will have a substantial impact on patient and professional healthcare education in the near future. A majority of the test users would at least partially recommend AR-supported tools for education. Furthermore, it was suggested that virtual training environments can promote user engagement and motivation and ensure reproducible training environments. In addition, potential cost and time savings supported by time- and space-independent training of AR educational tools were highlighted.

In general, this paper leads to interesting prospects for the use of AR in the education of radiographers, healthcare professionals, or in education in general. As of now, AR is still limited by its hardware. Headsets are expensive and intended for professional use rather than for personal use, thereby practically excluding studying at home from current use cases. With hardware becoming more accessible and capable, the possibilities of AR in education are nearly endless. Higher-fidelity holograms, better utilization of real-world objects, and more intuitive input methods would increase user immersion and acceptance of the technology, which leads to questions regarding the impact that AR applications can have on learning in general, and regarding how gamification could benefit virtual training applications.

## Figures and Tables

**Figure 1 healthcare-10-00672-f001:**
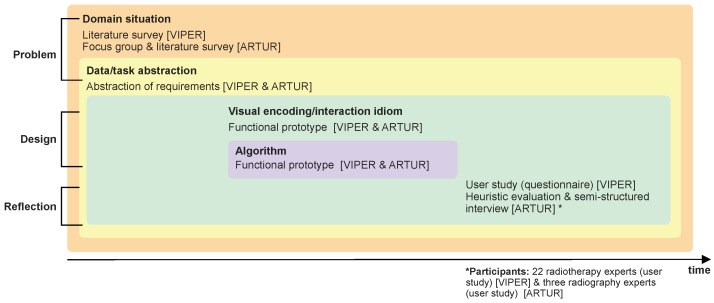
We followed a nested model [16], along with a design study methodology [15]. We started with the problem characterization, including the domain situation and data/task abstraction, followed by the design, with the visual encoding/interaction idiom and algorithm. Finally, we reflect on the results of the conducted user study.

**Figure 2 healthcare-10-00672-f002:**
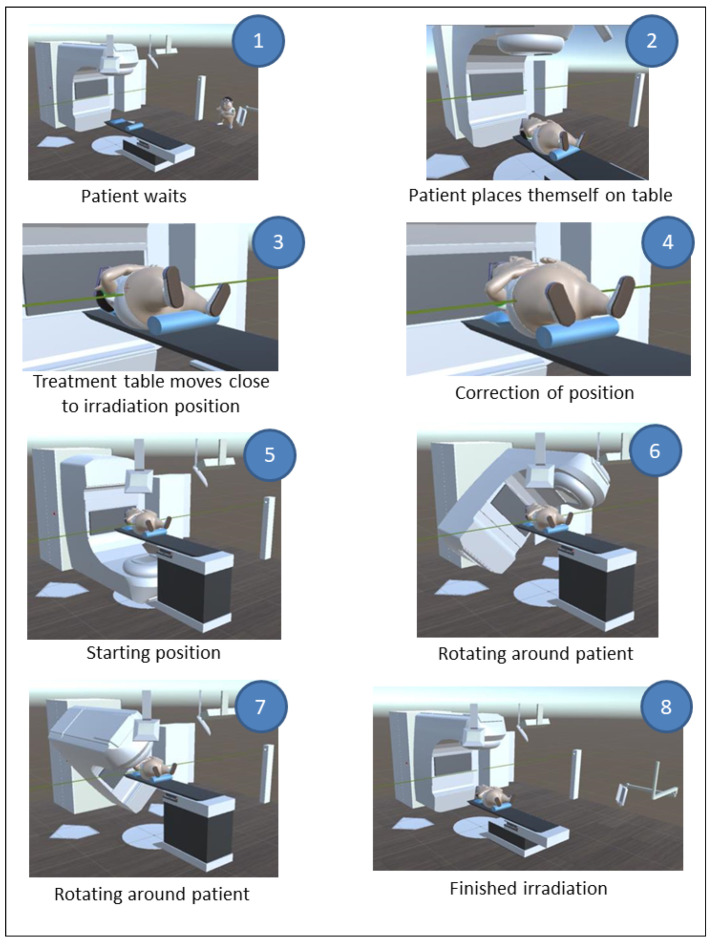
Sequence of the patient information of VIPER.

**Figure 3 healthcare-10-00672-f003:**
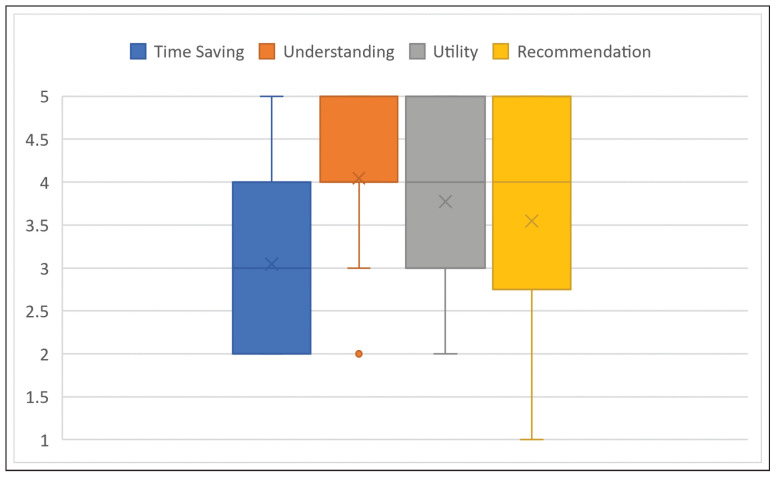
Expected influence of VIPER on increased understanding and time saving assessed by the participants and estimation of the potential of VIPER in terms of usefulness and recommendation potential when applied as a patient education tool. Likert scale from 1 to 5 (1: does not apply at all; 5: fully applies).

**Figure 4 healthcare-10-00672-f004:**
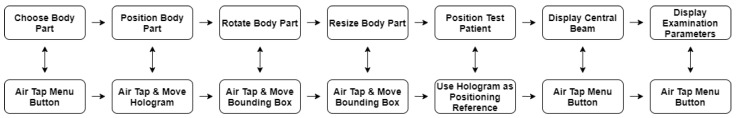
Subdivided user tasks with their corresponding user interactions. (1) Choose a body part; (2) position the body part on the training surface; (3) rotate the body part; (4) resize the hologram; (5) position the test patient; (6) display the central beam; (7) display additional parameters.

**Figure 5 healthcare-10-00672-f005:**
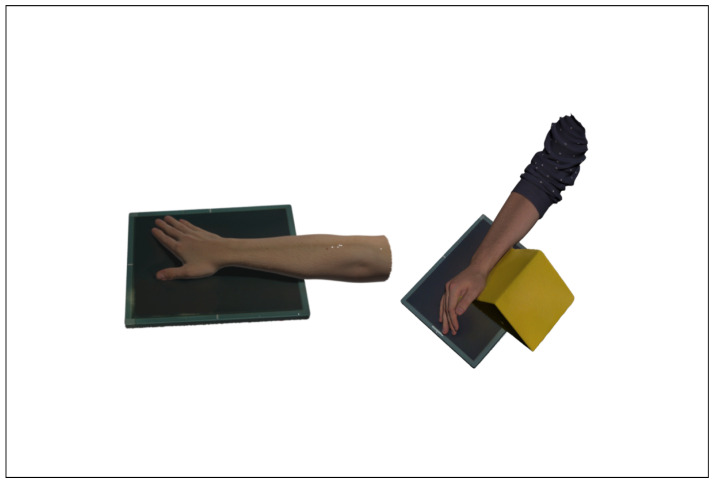
Three-dimensionally scanned arm in position for two radiographs of the scaphoid bone.

**Figure 6 healthcare-10-00672-f006:**
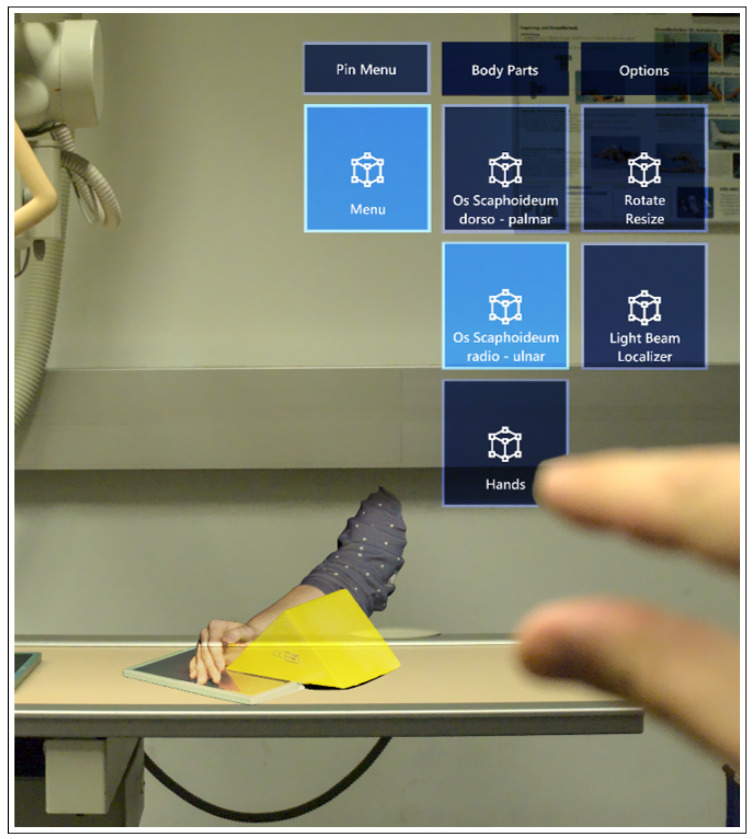
User interaction with ARTUR from the user’s FOV.

**Table 1 healthcare-10-00672-t001:** Identified usability issues of ARTUR and their severity rating. Lack of comfort, limited FOV, and limited interaction methods were linked to the use of Microsoft HoloLens 1 hardware and received a severity rating of zero. The similarity between UI buttons and labels was labeled as a cosmetic issue with a severity rating of one. Occasional collider occlusion was rated as a minor usability problem with a severity rating of two.

Issue	Severity Rating
Headset Comfort	0
Limited FOV	0
Limited Interaction Methods	0
No Distinction between Buttons and Labels	1
Collider Occlusion	2

## Data Availability

The data presented in this study are available in https://phaidra.fhstp.ac.at/o:4773, accessed on 30 January 2022.

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
