# Peer review of "Augmented Reality in Radiology for Education and Training—A Design Study"

_healthcare, 2022, doi:10.3390/healthcare10040672_

Round 1
Reviewer 1 Report
The present study proposes and evaluates VIPER and ARTUR AR applications for radiography and radiotherapy education and training. The manuscript is well organized and comprehensively described, but the references are missing. The information provided allows forming an overview on the proposed research. The methodology is good and the results of the two use cases are convincing. The results are helpful for education of radiographers and healthcare professionals. Some aspects of the paper must be improved: add the references; Pag 12: “Input Feedback: Haptic, visual and audible feedback” is confusing. Haptic, visual and audio are outputs not inputs.
Reviewer 2 Report
Radiology education and training are excellent research topics. This paper presents 2 promising prototypes for patient education and to train radiographer technician students. However, the manuscript presents serious flaws that make it unrecommendable for acceptance. First of all, the work is very badly motivated: authors present 2 interactive systems for radiology education and training that seem independent of each other. Why these two studies? It makes more sense to report 2 independent papers (1 for VIPER, 1 for ARTUR). Secondly, the user studies are too subjective and lack proper statistical analysis to be properly validated. A positive note is that 22 experts participated in the VIPER study. However, authors should have conducted a more in-depth and through user study (there is so much feedback that an expert can provide!). VIPER evaluation falls a bit too short given that 22 radiotherapy experts were invited to the study. If so many experts were available, then authors should haved conducted a more formal and richer study (note that there is no measurement of SUS, NASA-TLX, IPQ, etc, nor has a (semi-strucutured) inverview been conducted. Moreover, metrics such as 'time saving' and 'Increased Understanding' appear out of nowhere (Table 1), no mention on how such metrics were measured and no statistical analysis was considered whatsoever. Apparently these metrics result from experts subjective understanding ... they do not correspond to a solid/robust objective measurement. ARTUR was evaluated with just 3 experts which is insufficient.
Authors did not explain why VIPER lacks a "focus group treatment" and the evaluation of both VIPER and ARTUR are different (questionnaire was considered for VIPER but ARTUR considered an heuristic evaluation and a semi-structured interview)
In addition, the title, abstract (motivation sentences) and introduction (motivation paragraph (1st paragraph)) are too generic.
Minor comments:
- references appear in the text with [?]
- figure 1: L/literatur --> L/literature
- Reference section simply does not exist
- Missing reference: M. Sousa, D. Mendes, S. Paulo, N. Matela, J. Jorge, D.S. Lopes, VRRRRoom: Virtual Reality for radiologists in the reading room, Proceedings of the 35th Annual ACM Conference on Human Factors in Computing Systems (CHI 2017), New York: ACM Press, 2017. DOI: 10.1145/3025453.3025566
Reviewer 3 Report
The two applications described in this paper are interesting. However, some issues need to be addressed before the manuscript is published.
- All references are missing and it is therefore impossible to assess their relevance to the manuscript and the completeness of the literature cited.
- According to the author, the VIPER system is a “virtual patient education too that explains the exact process of irradiation” and “a requirement for the development of VIPER was to enable mobility, while at the same time ensuring easy handling by the users.”Moreover, from the application description, it seems that there is no interaction between the VR content and the real environment, and the HoloLens is simply used to display a VR animation. So why did you choose an AR visor rather than a VR visor? The authors should better clarify the choice.
- For the Likert questionnaire, it is better to summarize results in terms of median and IQR.
- It would be desirable to recruit more subjects for testing the ARTUR system.
- Line 584 “The issues identified were: lack of comfort while wearing the headset, the seriously limited FOV provided by the first-generation HoloLens, and the input methods limited to a few gestures only” This is absolutely true, could you add information about the next version of HoloLens and how it could partially fix the issues (improvement in the field of view, less weight, etc.).
Round 2
Reviewer 2 Report
Unfortunately, authors maintain their position and did not perform any major revision to the document. Instead, they do not provide any solid justification to my previous comments. The same issues persist, therefore, I cannot recommend this paper to be published in its current form.
Author Response
Dear Reviewer 2, thank you for your feedback. We added Proof of concept directly in the abstract! Furthermore, we also improve the English language. We instruct our internal department to proof-read our manuscript. Best regards, Christina Stoiber
